



# A multi-fluid model of the magnetopause

Roberto Manuzzo[1,2], Francesco Califano[2], Gerard Belmont[1], and Laurence Rezeau[1]

[1]LPP, CNRS, Ecole Polytechnique, Sorbonne Université, Univ. Paris-Sud, Observatoire de Paris, Université Paris-Saclay, PSL Research University
[2]Department of Physics E. Fermi, Universitá di Pisa, Italia

**Correspondence:** F. Califano (francesco.califano@unipi.it)

**Abstract.** Observation of the solar wind - magnetosphere boundary provides a unique opportunity to investigate the physics underlying the interaction between two collisionless magnetized plasmas with different temperature, density and magnetic field topology. Their mixing across the interface as well as the boundary dynamics are affected by the development of fluid (and kinetic) instabilities driven by large scale inhomogeneities in particle and electromagnetic fields. Building up a realistic initial

equilibrium state of the magnetopause according to observations is still a challenge nowadays. In this paper we address the modeling of the particles and electromagnetic fields configuration across the Earth's magnetopause by means of a three-fluid analytic model. The model relies on one hot and one cold ion population and on a neutralizing electron population. The goal is to build up an analytic model able to reproduce as closely as possible the observations. Some parameters of the model are set by using a fit procedure aiming at minimizing their difference with respect to experimental data provided by the Magnetospheric

MultiScale mission. All the other profiles, concerning the electron pressure and the relative densities of the cold and hot ion populations, are calculated in order to satisfy the fluid equilibrium equations. Finally, by means of a new tri-fluid code, we have checked the stability of the large-scale equilibrium model for a given experimental case and given the proof that the system is unstable to reconnection. This model could be of interest for the interpretation of satellite results and for the study of the dynamics at the boundary between the Magnetosphere and the solar wind.

## 1   Introduction

The solar wind - magnetosphere boundary, known as the magnetopause, is characterized by the presence of magnetic and velocity shears as well as jumps in magnetic and velocity magnitudes, in density and temperature. These inhomogeneities are the sources of many plasma instabilities at different spatio-temporal scales (Labelle and Treumann, 1988), in turn often triggering

themselves secondary instabilities at smaller scales. As an example, secondary instabilities such as magnetic reconnection, Kelvin-Helmholtz and/or Rayleigh-Taylor instability can efficiently develop on the shoulder of the primary instability as for instance the Kelvin-Helmholtz at the low latitude magnetopause (see Faganello and Califano (2017) and references therein).


All of these phenomena can cause significant entering of magnetosheath plasma mass (Paschmann, 1997), momentum (Dungey, 1961) and energy (Lee and Roederer, 1982) into the magnetosphere. The study of the magnetopause is of partic-
ular interest since this system offers the unique opportunity to study a two-plasma large-scale interaction in conditions not achievable in laboratory. The magnetopause physics is also of basic importance in the studies addressing the Sun-Earth in-teraction, in particular concerning the impact of solar wind disturbances on the terrestrial environment and the attempts of space-weather forecasting (see for instance Baker and Lanzerotti (2016)). The question of modelling space plasmas using data provided by multi-spacecraft missions has been much developed during the Cluster era (Büchner et al., 1998). Concerning the
magnetopause data, one of the key points concerns the mixing between magnetospheric and magnetosheath plasmas and the resulting non-Maxwellian shape of the distribution functions (hereafter d.f.) observed in these regions (Bosqued et al. (2001), Frey et al. (2003), Phan et al. (2005), Retinò et al. (2005)). These d.f. are often reminiscent of those observed in reconnec-tion kinetic simulations (Nakamura and Scholer (2000), Tanaka et al. (2008), Aunai et al. (2011)). Some of these d.f. can be compared with simple analytic models as in the pioneering work by Cowley and Owen (1989). Since the populations coming
from the two different sides of the magnetopause differ in density and in temperature, modeling the mixing requires at least the use of a multi-population model. In the perspective of investigating the dynamics of the magnetopause mixing layer by a three-fluid numerical simulation, the main target of this paper is to build up a three-fluid equilibrium as realistic as possible for initializing it.

Several multi-population models trying to simulate the plasma exchanges between magnetosheath and magnetosphere have
been developed in the past. In particular kinetic models must provide a Vlasov equilibrium. Such models are very complicated so that the authors are lead to make simplistic mathematical assumptions for choosing the initial d.f., hopeless to get close to magnetopause observed profiles, as for instance the velocity and/or the magnetic field ones. Furthermore, these models involve many free parameters even in the simplest limit of a plane and tangential layer (*i.e.* without a normal magnetic field: $B_n = 0$). There is no constraint, in particular for fixing the initial electric field profile of a tangential discontinuity in these approaches.
Note also that all the equilibria built via d.f. that are functions of the particle invariants of motion only (Channell (1976)) cannot really be considered as "multi-population" models: they ignore the questions of accessibility and they can therefore not distinguish between particles of magnetospheric or magnetosheath origin. Some recent models (see Belmont et al. (2012) and Dorville et al. (2015) and references therein) allow to solve this problem only in part. Indeed in these models even if a few profiles can be fixed in a, let say, realistic way, all the other instead still depend on simple mathematical assumptions, which
are largely arbitrary, so that they are still far from realistic.

In summary, the lack of realistic equilibria in the literature makes difficult, for kinetic simulations, the initialization of the magnetopause studies. Nevertheless, the multi-population character of the medium has been taken into account in a recent paper (Dargent et al., 2017) addressing the influence of cold and hot magnetospheric ions on the development of magnetic reconnection. In this paper, the magnetospheric plasma includes two populations with different temperatures in order to account
for the presence of cold ions in the magnetosphere close to the magnetopause.

Multi-fluid models have been developed in various domains, but in general not for magnetopause studies. These studies address multi-species evolution involving chemical processes and collisions. They have been used to investigate planetary





atmospheres (Modolo et al. (2006), Ma et al. (2007)), the solar chromosphere (Alvarez Laguna et al., 2016), basic plasma physics problems (drift turbulence in Shumlak et al. (2011) for instance).

In this paper we present a new technique to build up a three fluid equilibrium that derives directly from satellite observations. The model assumes uni-dimensional gradients in the normal direction and a tangential boundary ($B_n = 0$) at the magnetopause. The magnetic and velocity shear are both taken into account in a realistic way. The profiles are chosen to fit at best data from the Magnetospheric MultiScale mission (MMS) (Burch et al., 2016b) for which the time-to-space conversion has been performed by means of recent techniques presented in (Manuzzo et al., 2019, under review). As it will be shown in section (4), the method

provides a cold and a hot contributions in qualitative agreement with observations, even if the model uses, as inputs, only the global ion macroscopic moments.

## 2   Observations

We use MMS data during the period October 16th 2015, 13:05:34 + 40s UT, which embeds a magnetopause crossing. In Figure (1) we plot the experimental data that the equilibrium model attempts to reproduce. This interval shows the standard signatures

of the region where magnetospheric and magnetosheath plasmas meet (magnetopause crossing): reversal of the magnetic field and change in the energy distributions.

    In panels (a), (b) and (c) data are plotted as functions of a spatial coordinate $X_n = X_n(t)$ which is the projection of the spacecraft path along the direction normal to the magnetopause (units of $d_{i,MSh}$ with $d_{i,MSh} \simeq 70km$). $X_n(t)$ is obtained from the temporal integration of the magnetopause magnetic structure velocity by means of a combination of three distinct

methods, STD+, SVF and MVF, optimized with a technique explained in a recent work presently under review on JGR (Manuzzo et al., submitted). Assuming a quasi-stationary structure for the magnetopause current sheet, the procedure gives the position of the probed data with respect to this structure. The main point of this technique is to allow one to recover the spatial profiles of quantities of interest when crossing the magnetosheath - magnetosphere boundary with a variable velocity. For the sake of completeness, we give also in the abscissa of panel (c) the time corresponding to each given value of $X_n(t)$.

In the two spectrograms (panels b and c), black points have been over-plotted to indicate their maxima. This allows one to individuate more easily where the magnetosheath and the magnetospheric plasma interact, as indicated by discontinuities in the curve joining the maxima. The mixing region, emphasised by a blue rectangle in panel (b), is located at $X_n \sim 3d_i$.

    In panels (d), (e) and (f) we plot the 2D ion distribution functions (i.d.f.) in the plane tangential to the magnetopause. They are obtained by integration over the out-of-plane (normal) component of the velocity. Each plot is the average of 5 single i.d.f

recorded within a $\sim 0.75s$ long interval (equivalent to $0.5d_i$). The radius of the distribution functions is $10^3$ km/s and the purple full circle drawn at their centres determines the bottom limits in energy of the FPI instrument (10 eV $\sim$ 53 km/s for ions). The direction of the local magnetic field is indicated by a white arrow. In panel (e) (mixing region), one can observe that the i.d.f. contains two peaks emphasised by the over-plotted circles (blue and red dashed lines). These two circles have a diameter equal to the magnetosheath and magnetospheric thermal velocities, respectively. The same circles are shown for

the magnetosheath and magnetospheric i.d.f.s, (f) and (d) panels. In these two asymptotic media, we see that there is only



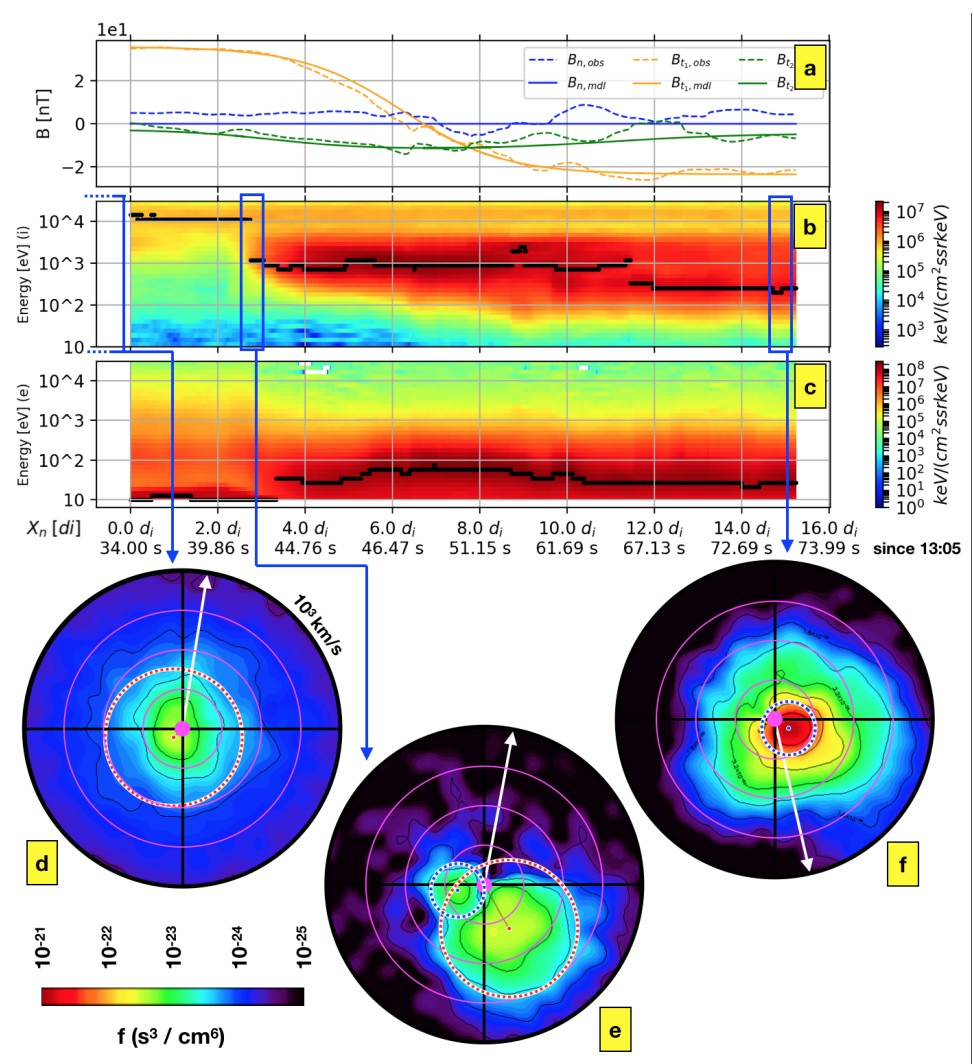

**Figure 1.** MMS data for the October 16th 2015, 13:05:34 UT + 40s event. Panel $a$: normal and tangential (to the magnetopause plane) components of the magnetic field. Panels $b$ and $c$: ion and electron spectrograms. The first abscissa is the spatial coordinate normal to the magnetopause $X_n$ (see text) and the second one is time. Panels $d$, $e$ and $f$: ion distribution functions recorded by the FPI instruments, respectively in the magnetosphere, in the overlapping region and in the magnetosheath. These i.d.f.s are projected on the tangential plane by integration over the normal component of the velocity.





one single peak. Note that the i.d.f. shown in panel (d) has been recorded a little earlier (10:20:00 UT + 2s) during a "clear" observation of the magnetosphere allowing to avoid the presence of magnetosheath particles when the spacecraft is too close to the magnetopause. On the other hand panel (e) shows a mixture of the magnetosheath and magnetospheric populations at the same time. However, since the two peaks are close to each other and since the distributions of the two populations are partly

superposed, it is not possible to clearly separate the hot/cold contributions, a necessary input for the multi-population model to be built by a direct fit of this region. We will explain in the next section a new method capable of separating the two particle components even in such complex situations. Note that, in the magnetospheric region ($X_n \leq 2d_i$) the electron spectrogram of panel (c) shows energy maxima that lay just at the bottom limit of the instrument ($\simeq 10\ eV$). This indicates the presence of cold electrons in the magnetosphere. The role of this poorly measured cold electron population is not relevant for the magnetopause

pressure equilibrium, but in the electron bulk velocity it could be significant. However the electron population parameters will be not determined by a direct fit of the data, nor will those of the two ion populations (cold/hot). They will be determined instead by another method based on the equilibrium equations, which we will describe in the next section.

## 3  The three fluid model

### 3.1  Equilibrium equations

We present here a 3fluid collisionless model which includes two ion populations (one cold and one hot) and one electron population. The cold ion population models the ions of magnetosheath origin and disappears more and more on the magnetospheric side. Conversely, the hot population models the ions of magnetosphere origin and disappears on the magnetosheath side.

The continuity and ion momentum equations are derived from the first two moments of the Vlasov equation. We impose charge neutrality and the displacement current is neglected. We assume isotropic pressures and adopt a polytropic closure for

all populations. These equations are coupled to the electromagnetic fields via the Faraday's equation and we use an Ohm's law taking into account the electron pressure gradient but neglecting electron inertial effects.

We normalise the 3fluid set of equations by using ion quantities, the proton mass and charge $m_p$ and $e$, respectively, the ion cyclotron frequency $\Omega_{ci} = e\bar{B}/m_p c$, the ion inertial length $d_i = c/\omega_{pi}$ where $\omega_{pi} = (4\pi\bar{n}m_p/e)^{1/2}$. In particular the characteristic density $\bar{n}$ and mean magnetic field $\bar{B}$ are taken in the magnetosheath far from the central inhomogeneous region. Using

index $\alpha$ when all plasma populations are concerned and $\beta$ when only the ion populations are concerned, the non dimensional 3fluid system of equations reads:



$$\begin{cases} \sum_\alpha sign(q_\alpha)n_\alpha = 0 & \text{(1a)} \\[2mm] \sum_\alpha sign(q_\alpha)n_\alpha \mathbf{U}_\alpha = \overline{\nabla} \times \mathbf{B} & \text{(1b)} \\[2mm] \dfrac{\partial n_\beta}{\partial t} + \overline{\nabla} \cdot (n_\beta \mathbf{U}_\beta) = 0 & \text{(1c)} \\[2mm] \dfrac{\partial (n_\beta \mathbf{U}_\beta)}{\partial t} + \overline{\nabla} \cdot (n_\beta \mathbf{U}_\beta \mathbf{U}_\beta) + \overline{\nabla}(n_\beta T_\beta) = n_\beta(\mathbf{E} + \mathbf{U}_\beta \times \mathbf{B}) & \text{(1d)} \\[2mm] \dfrac{\partial (n_\alpha S_\alpha)}{\partial t} + \overline{\nabla} \cdot [\mathbf{U}_\alpha(n_\alpha S_\alpha)] = 0 \text{ with } S_\alpha = T_\alpha n_\alpha^{1-\gamma} & \text{(1e)} \\[2mm] \dfrac{\partial \mathbf{B}}{\partial t} = -\overline{\nabla} \times \mathbf{E} & \text{(1f)} \\[2mm] \mathbf{E} = -(\mathbf{U}_e \times \mathbf{B} + \dfrac{1}{n_e}\overline{\nabla}(n_e T_e)) & \text{(1g)} \end{cases}$$

## 3.2 Determination of the fluid profiles

We aim at establishing a tangential 1D equilibrium to mimic as close as possible the magnetopause observations previously
presented. Assuming $\partial/\partial t = 0$, this is done in three steps.

Step 1: We impose the magnetic field $\mathbf{B}$, the density $n_i$, temperature $T_i$ and velocity $\mathbf{U}_i$ profiles these last without distin-
guishing the cold and hot populations. This is done by fitting the data using a combination of hyperbolic tangents as explained
in section (4).

Step 2: We deduce the electron density $n_e$ and velocity $\mathbf{U}_e$ by using the equilibrium equations (1a) and (1b). The temperature
$T_e$ is deduced from

$$P_e = P_{tot} - (P_B + n_i T_i) \tag{2}$$

where the total pressure, $P_{tot}$, is assumed to be a constant in order to fulfil the equilibrium conditions.

As far as $P_e$ is concerned, we note that $i)$ $P_e$ is much smaller than $P_i + P_B$ (see Fig. 2) and that $ii)$ it is difficult to estimate
it precisely because of experimental uncertainties. As a consequence, we take for $P_{tot}$ the maximum of the measured total
pressure $P_i + P_B + P_e$ and we deduce the modeled $P_e$ from equation 2; this ensures one to get only positive values for $P_e$.

Finally, the electric field $\mathbf{E}$ is deduced from the Ohm's Law, Equation (1g).

Step 3: We now split the global proton population into two different populations, cold and hot (hereafter "$ic$" and "$ih$",
respectively) to distinguish the magnetospheric and magnetosheath populations. The densities ($n_{ic}$ and $n_{ih}$), pressures ($P_{ic}$
and $P_{ih}$) and currents ($\mathbf{J}_{ic}$ and $\mathbf{J}_{ih}$) of the two ion populations add to form the total ion density, pressure and current as follows





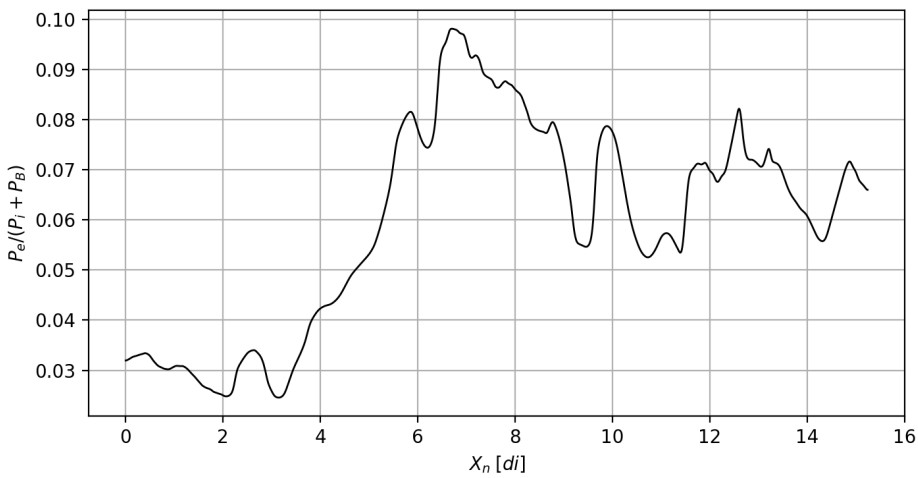

**Figure 2.** Comparison between $P_e$ and the other pressure terms $P_i$ and $P_B$. $P_e$ is small everywhere, both within the magnetosphere ($X_n \leq 2.0 d_i$) and in the magnetosheath ($X_n \geq 12.0 d_i$).

$$n_i = n_{ic} + n_{ih} \tag{3a}$$

$$n_i T_i = n_{ic} T_{ic} + n_{ih} T_{ih} \tag{3b}$$

$$n_i \mathbf{U}_i = n_{ic} \mathbf{U}_{ic} + n_{ih} \mathbf{U}_{ih} \tag{3c}$$


The temperatures of the cold and hot ion populations, $T_{ic}$ and $T_{ih}$, are assumed to be constant. Since the global ion temperature profile $T_i$ is known, their values are obtained from the satellite data by the two limits:

$$\lim_{x \to MSph} T_{ih} = T_i ; \quad \lim_{x \to MSh} T_{ic} = T_i$$

The temperature ratio between the two populations is set by the value of the dimensionless parameter:

$$\Upsilon \equiv \frac{T_{ih}}{T_{ic}} \tag{4}$$

Using Equation (3b), the contributions of each population to density and pressure are fully determined by the $T_i$ profile and the temperature ratio $\Upsilon$:

$$\Gamma \equiv \frac{n_{ic}}{n_i} = \frac{\Upsilon - \frac{T_i}{T_{ic}}}{\Upsilon - 1} \tag{5a}$$

$$\Pi \equiv \frac{P_{ic}}{P_i} = \left( 1 + \frac{1 - \Gamma}{\Gamma} \Upsilon \right)^{-1} \tag{5b}$$



The perpendicular currents and by consequence the corresponding velocities, are fully determined by Equations (1d). On the

contrary, the parallel currents cannot be determined by the above system of equilibrium equations. We will set them by a

reasonable choice for the parameter $\phi$ which is equal to the ratio of the cold parallel ion current to the total parallel ion current

as seen in the electrons frame:

$$\phi \equiv \Gamma \frac{(\mathbf{U}_{ic} - \mathbf{U}_e) \cdot \mathbf{b}}{(\mathbf{U}_i - \mathbf{U}_e) \cdot \mathbf{b}} \tag{6}$$

The parallel components of the hot and cold ion velocities can have opposite directions, so that $\phi$ is defined in the $[-1, 1]$ range,

while $\Gamma$ and $\Pi$ are defined in the $[0, 1]$ range. The reasonable choice for $\phi$ is suggested by the data and will be discussed in

more details in the next section. In general, the asymptotic values of the cold and hot ion currents are chosen in agreement with

the asymptotic values of $n_{ic}$ and $n_{ih}$, in order that all the corresponding values of the velocities $U_{ic}$ and $U_{ih}$ have reasonable

values, although one of the two densities $n_{ic}$ or $n_{ih}$ tends to nearly zero on each side.

In order to implement this model into a numerical simulation, a compromise is necessary since the multi-fluid code cannot

deal with a population having a zero density somewhere in the domain. To avoid this problem, we introduce the parameters

$\epsilon^{(c)} \ll 1$ and $\epsilon^{(h)} \ll 1$ and we modify the initialisation so that the cold and hot densities tend to $\epsilon^{(h)} n_i$ and $\left(1 - \epsilon^{(h)}\right) n_i$ on

the magnetospheric side, and vice versa to $\left(1 - \epsilon^{(c)}\right) n_i$ and $\epsilon^{(c)} n_i$ on the magnetosheath side. The temperatures are changed

according to:

$$T_{ic} = \frac{\epsilon^{(c)} T_i^{MSph} - \left(1 - \epsilon^{(h)}\right) T_i^{MSh}}{\epsilon^{(c)} + \epsilon^{(h)} - 1} \tag{7a}$$

$$T_{ih} = \frac{\epsilon^{(h)} T_i^{MSh} - \left(1 - \epsilon^{(c)}\right) T_i^{MSph}}{\epsilon^{(c)} + \epsilon^{(h)} - 1} \tag{7b}$$

where $T_{ic}$ and $T_{ih}$ indicate the observed values corresponding to the model, and $T_i^{MSph}$ and $T_i^{MSh}$ the temperatures corre-

sponding to the magnetospheric and magnetosheath values of $T_i$. A similar correction is made for the ion velocities (see next

sections).

## 4  Data vs analytic profiles

We apply the procedure to the case study introduced in section (2). In Figure (3) we compare the model field profiles with the

ones obtained with the MMS data. The model profiles for the magnetic field, the ion temperature and density are obtained by a

fit procedure, panels (a), (b), and (c), respectively, while the others are calculated from the equilibrium equations. The fits are

obtained by means of analytic functions. For a given quantity $Q$, the fitting functions have the following form:

$$Q = \sum_j a_{Q,j} + b_{Q,j} tanh(\frac{X_n - c_{Q,j}}{d_{Q,j}}) \tag{8}$$

where $X_n$ is the coordinate along the direction normal to the magnetopause (as discussed in section 2). The parameters $a_{Q,j}$,

$b_{Q,j}$, $c_{Q,j}$, and $d_{Q,j}$ are the free parameters shaping the analytic profiles and $j$ is the component index. The maximum value





of $j$ depends on the fitted quantity: the analytic profiles are considered as good fits of the data if they correctly shape the large scale configuration, as well as the position and the length scale of the gradients within the magnetopause layer. An example of such a "good fit" is given in Fig. 3, panels (a), (b) and (c). It is worth noticing that the particle boundary, observed on density and temperature, has a length scale smaller than the magnetic boundary (by a ratio $\simeq 0.25$) and that its position is considerably

shifted toward the magnetosphere with respect to the centre of the magnetic jump. This may indicate the presence of a boundary layer, possibly made of magnetosheath plasma observed on the magnetospheric side of the magnetopause (Hasegawa, 2012). Such features cannot be reproduced in the framework of a MHD equilibrium model. To the best of our knowledge, they have not been introduced even in the context of a kinetic model.

In panel (b) we show the temperature profiles as obtained with our model equilibrium. The total ion population temperature

$T_i$ has been obtained by fit, and it is superposed with the cold ion population temperature $T_{ic}$ (blue curve) and the hot ion population one $T_{ih}$ (red curve). The figure has been drawn using $\epsilon^{(h)} = 0.35$ and $\epsilon^{(c)} = 0.05$, which determines the values of $T_{ic}$ and $T_{ih}$ via Eqs.(7).

One observes that the global temperature is well fitted by the model outside the mixing region, but that the fit is less accurate in the $\sim 1.0d_i \leq X_n \leq\sim 2.5d_i$ interval. In this interval the real total ion temperature becomes actually larger than its

magnetospheric asymptotic limit. Unfortunately this feature can not be reproduced by the present 3fluid model with constant hot and cold temperatures since the $\Gamma \geq 0$ constraint forces the $T_i$ profile to be everywhere lower than $T_{ih}$ (see Equation (5a)). This little deviation is acceptable since the model mainly aims at reproducing the asymptotic trends, the observed inner region probably being out of equilibrium.

In panel (c) we show the density profiles. As explained in the previous section, the hot and cold ion contribution to the total

density $n_i$ are computed by means of the $\Gamma$ function which is fixed once the global $T_i$ profile and the temperature ratio $\Upsilon$ are fixed (Equation (5a)). For all panels of Figure (3) the two vertical lines (dashed black) indicate the limits of the region where $1/4 \leq \Gamma \leq 3/4$. Note that the cold ion density falls rapidly to very low values in more or less $\sim 2\,d_i$ while the hot population density keeps nearly the same value over a longer interval (between 0 and 8 $d_i$).

The electron density and velocity profiles are obtained from the equilibrium equations. However these quantities are not

plotted here since their experimental counterparts are likely to be biased in the magnetosphere by the cold electron population which is below the bottom threshold in energy of the FPI instrument (as mentioned in section 2). On the other hand, we plot in panel (d) the electric field, which is obtained from the 3fluid model, Equation (1g). We see that the electric field calculated by the model agrees quite well with the one measured by the instrument independently of the electron measurements.

The parallel components of the cold and hot ion currents are set by $\phi$ (Equation 6). As long as there are no cold ions on the

magnetospheric side and no hot ions on the magnetosheath side, the asymptotic constraints on $\phi$ would be

$$\lim_{x \to MSph} \phi = 0 \,; \quad \lim_{x \to MSh} \phi = 1$$

Nevertheless, because of the compromise necessary for implementing the model in the multi-population numerical simulation, the cold and hot densities actually take small but not strictly null values on both sides. To determine the corresponding parallel currents, corrections similar to Eqs. (7) are applied with the assumption that, on each side, the parallel velocities of the cold

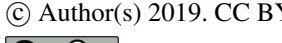



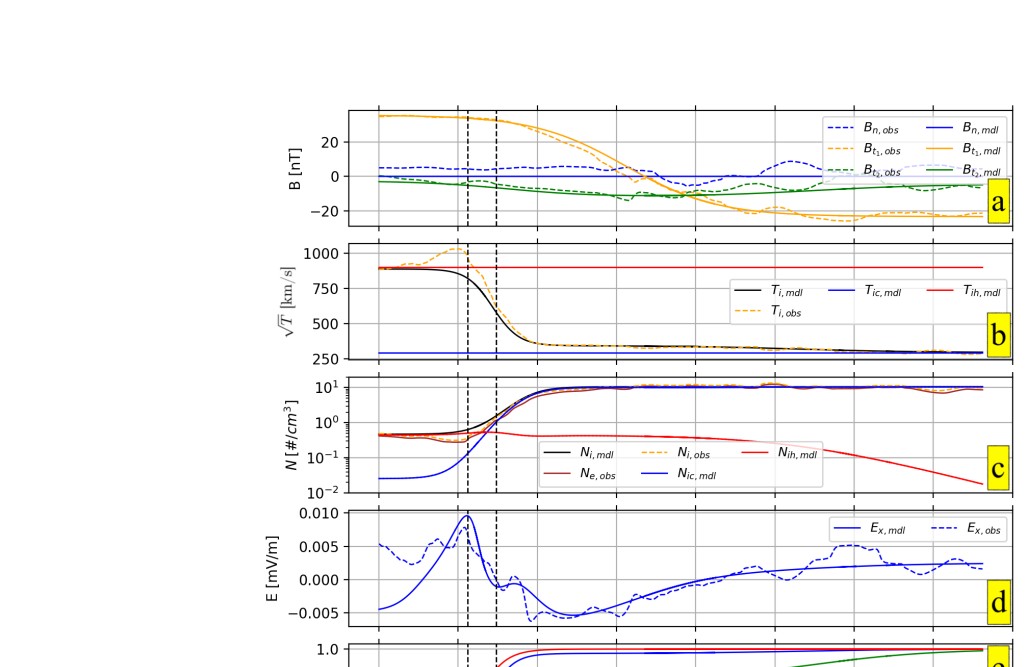

**Figure 3.** Comparison between the magnetopause profiles as observed by MMS during the 16 October 2015, 13:05:34 + 40s UT period and those used for the 3fluid model equilibrium. Satellite data are represented by dashed lines, the extrapolated profiles used in the model by continuous lines. The $X_n$ coordinate represents the spatial coordinate normal to the magnetopause $X_n$. In the panels we show the magnetic field (a), the temperatures (b), the density (c), the electric field (d), the parameter $\Gamma$, $\Pi$ and $\Phi$ (e), and the parallel and perpendicular components of the ion current (panels f and g respectively). The two vertical lines (black dashed) highlight the transition region ($1/4 \leq \Gamma \leq 3.4$). The blue and orange colours adopted for the electric and magnetic fields represent the normal and the tangential components of the fields. The square roots of the temperatures (panel b) are plotted in velocity units in order to make easier the comparison with i.d.f.s shown in Figure (1). For the sake of clarity, the curves shown in panels (f) and (g) have been multiplied by a factor 10 in the $0.0 \leq X_n \leq 3.0$ interval.





and hot populations, in the electron frame, are equal to each other and therefore equal to the global one. Under this assumption, it can be easily shown that the asymptotic values of $\Phi$ are equal to those of $\Gamma$:

$$\lim_{x \to MSph} \phi = \epsilon^{(h)} ,; \quad \lim_{x \to MSh} \phi = 1 - \epsilon^{(c)},$$

Note that for the particular MMS event considered, the global ion parallel velocities are observed to be quasi-null on each side, so that the same asymptotic property holds for the velocities of the two populations.

Between the two limits above, a reasonable choice for the $\phi$ profile is that the length of its gradients be of the same order as the scale length of the density and temperature gradients, $i.e. \sim 1-2d_i$. The position of the main gradient of $\phi$ is set in order to separate the magnetopause thickness in two parts, each of length proportional to the gyro-radii of the two populations (their ratio is $\simeq 2$).

In panel (e) of Figure (3) we show the model profiles for $\Gamma$, $\Pi$ and $\Phi$. Because of the differences of temperature between the two components, the profile in $\Pi$ (concerning the pressures) noticeably differs from the profile in $\Gamma$ (concerning the densities).

Finally, in panels (f) and (g) of Figure (3), we show the results concerning the parallel and perpendicular components of the ion currents. Once more, one observe that the global ion current is well fitted, at the exception of the perpendicular current in the mixing region, which is less accurate. This is due to the small inaccuracy already mentioned of the modeled ion temperature in this region.

## 5 Numerical simulations

### 5.1 Set up

Here we give an example of a 3fluid numerical simulation aiming at demonstrating the possibility of studying numerically the above system starting from an equilibrium not far from a real one, not only qualitatively but also quantitatively. A detailed numerical study relying on such approach will be the focus of future work.

The three-fluid model introduced in this paper has been used to initialize a 2D 3fluid numerical simulation of the interaction between the solar wind and the Earth's magnetopause. The numerical simulation is intended to mimic the October 16th 2015, 13:05:34 + 40s UT MMS crossing. This simulation has been performed by using a 3fluid numerical code that solves the set of Equations (1a-g). The code originates from a 2fluid 3D parallel code largely used for the study of the interaction of the solar wind with the Magnetosphere (see Fadanelli et al. (2018) and references therein). The 3fluid code adapts the new equations to the algorithm of the 2fluid code presented in Faganello et al. (2009). It advances in time with a standard third-order Runge-Kutta algorithm (Canuto, 1988). It uses sixth order explicit finite differences along the periodic $y$ and $z$-direction and a sixth-order compact finite difference scheme with spectral like resolution for spatial derivative along the inhomogeneous $x$-direction. The numerical stability is guaranteed by means of a spectral filter along the periodic $y$ and $z$ directions and a spectral-like filtering scheme along the inhomogeneous $x$-direction. The code is parallelized along the periodic $y$ and $z$ directions (Lele, 1992). The code has been validated by standard numerical tests. In particular, by selecting separately the two cold and hot ion populations, we have reproduced the propagation of ion acoustic and Alfvén waves.





To initialize the simulation presented in this paper we take as initial equilibrium the model profiles represented in Figure (3), including the few modifications for the cold and hot ion density components (with respect to the basic model) because of the computational reasons discussed at the end of Section 3.

The simulation box dimensions are given by $L_x = 160d_i$, $L_y = 20\pi d_i$ and the box is discretized using $n_x = 800$ and $n_y = 320$ grid points corresponding to $dx = dy = 0.2d_i$. We have checked that the equilibrium configuration remains stable for several thousands of ion cyclotron times in the absence of an initial perturbation because of the very low values of the numerical noise and of the high accuracy of the numerical methods.

## 5.2 Results

The large scale equilibrium configuration used to initialize the simulation is unstable with respect to the reconnection mode. At $t = 0$ we add to the equilibrium an initial perturbation $\delta\mathbf{B} = \nabla \times \mathbf{A}$. The potential vector is given by a sum of random phase modes as follows:

$$A_l = \epsilon(x)\sum_{k_y}\sum_{k_x}\{cos\left[k_x x + k_y y + \phi_{1,l}(k_x, k_y)\right]+$$

$$cos\left[k_x x - k_y y + \phi_{2,l}(k_x, k_y)\right]\}/k; \quad l = x, y, z; \quad k = \sqrt{k_x^2 + k_y^2}; \quad i \neq j \tag{9}$$

where $\phi \in [0, 2\pi)$ are random phases and $\epsilon(x)$ is a Gaussian-like convolution profile in the inhomogeneous direction going to zero at both boundaries given by

$$\epsilon(x) = \epsilon_0 e^{-\left(\frac{x - x_{mp}}{2L_{mp}}\right)^2} \tag{10}$$

where $x_{mp} = L_x/2 = 60$ and $L_{mp} = 1.66$ are the position and the thickness of our magnetopause model.

The simulation is run for about 1500 ion cyclotron times. Very rapidly the initial perturbation reorganizes and sets up the reconnection eigenmodes that are identified by their wave-number in the $y$-periodic direction (each $m$ wave-number is easily recovered by taking the Fourier Transform of the perturbation along the $y$-direction at a given time). Following the classical reconnection theory (Furth et al., 1963) (but ignoring the density inhomogeneity), we have checked that our equilibrium is $\Delta'$ unstable for the first five eigenmodes. We recall here that the $\Delta'$ parameter depends on the equilibrium magnetic shear

and on the wavelength of the perturbation. It defines the instability threshold condition ($\Delta' \geq 0$). The unstable modes can be seen in Figure (4) where we plot $\Delta'$ as a function of the wave numbers $m_y$. We see that only the first five modes have a positive $\Delta'$, in agreement with the simulation where in the linear phase the $m_y \geq 6$ are stable (see discussion below). In Figure (5), panel (a), we plot the profile of the fastest growing eigenmode (corresponding to $m = 2$) of the $x$-component of the magnetic field fluctuation $\delta b_x$. The plot is along the inhomogeneous $x$-direction at five different times (see the legend) in log

scale. The two red vertical dashed lines indicate the spatial window of the equilibrium represented in Figure (3). This picture shows that after an initial transient needed to set up the normal mode shape, the reconnection instability develops around the





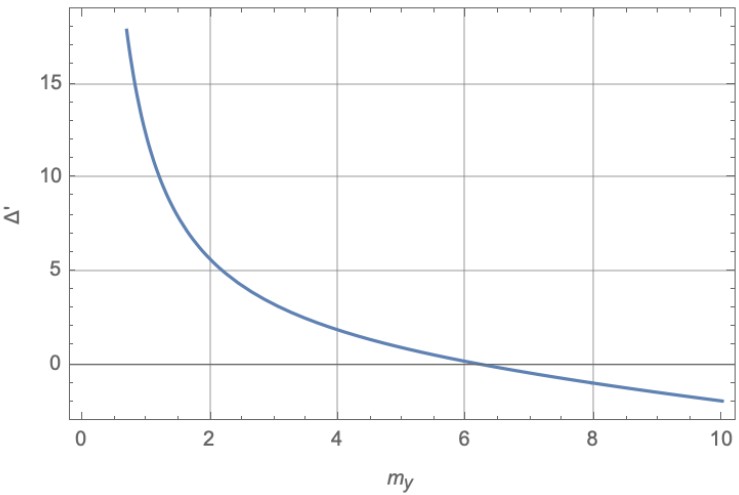

**Figure 4.** $\Delta'$ as a function of the wave numbers $m_y$ for an equilibrium magnetic field $\sim tanh(x)$.

region where the magnetic field reverts, $0 \leq x \leq 16$ (see also Figure (3)). Since the equilibrium is asymmetric, in particular for what concerns the cold and hot ion density that vary in a different location with respect to the point where the magnetic field inverts, the eigenmode is not symmetric with respect to the point where the magnetic field inverts, $X_n \simeq 6.4$. To the best of

our knowledge, this is the first time that one investigates the reconnection instability in the framework of a 3fluid approach in a non symmetric equilibrium representing directly the large-scale configuration taken from a satellite data event. Our goal here is to show the possibility to set up such a "realistic" large-scale initial equilibrium configuration to be simulated by a three-fluid approach. The non linear development of the system and in particular the mixing efficiency will be the object of future work. Still in Figure (5), panel (b), we plot the eigenmodes growth $vs$ time in normalized units (log scale). We see the exponential

growth of the first five modes, $m_y \equiv k_y L_y = 1,..,5$. Modes with $m_y = 6,7$ are instead stable. The orange curve corresponds to the most unstable mode, $m_y = 2$, the one plotted in panel (a). Despite the strong inhomogeneity of the system where, as discussed before, the magnetic inversion and the density variations occur at different locations, we see a very clear exponential growth with a constant growth rate. The linear phase last until about $t \simeq 1000$ after which the non linear phase begins. The values of the growth rates of the five unstable modes are reported in panel (c) confirming that $m_y = 2$ is the most unstable one.

In Figure (6) we show at the beginning of the saturated phase, $t = 1455$ the shaded iso-contours of the cold ion population, $N_{i,c}$. We see the formation of a hole structure corresponding to the region where the cold ion density grows eventually reaching the asymptotic magnetosheath value. To show the cold density hole, we made a cut along the inhomogeneous $x$-direction at $y = 38$ (see dashed line) still in Figure (6) in the bottom frame. In Figure (7) we show the same quantities for the hot ion fluctuations. We see a "complementary" behavior in the sense that now a bump is generated more or less in correspondence of

the cold ion hole. However, as already discussed, it is not the goal of this paper to study the non linear dynamics and mixing properties of the cold and hot ion populations. Our aim here is limited at presenting a method able to obtain a "realistic" 3fluid



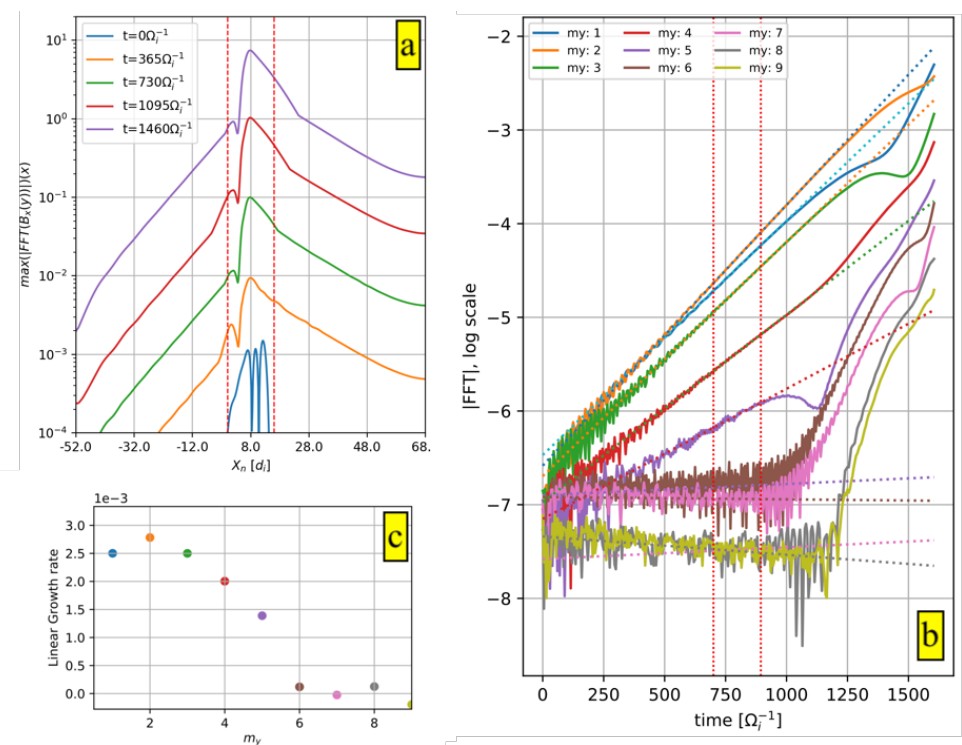

**Figure 5.** Development of the reconnection instability. Panel (a): the modulus of the Fourier Transform of $\delta b_x$ along $y$ $vs$ $x$ at five fixed time instant. The plots correspond to the fastest growing mode, $m = 2$, in log scale. The two red dashed vertical lines indicate the space interval of Figure (3). Panel (b): the first five eigenmodes growth $vs$ time. The orange curve corresponds to the most unstable mode, $m = 2$, the one plotted in panel (a). Panel (c): the growth rate values $vs$ $k_y$ calculated by a best fit of the slopes in panel $b$. The colors correspond to those used in panel (b).

equilibrium starting from a set of satellite data that can be used as initial condition for the investigation of the dynamics in the framework of a three-fluid approach.

## 6   Conclusions

The huge amount of spacecraft data today available brings a lot of information about the magnetopause, especially those of the MMS mission with their high resolution particle data. The magnetopause modeling can now be improved in view of these observations, which show that this boundary is never the simple textbook boundary generally considered. Beyond the natural asymmetry in temperature and density between the magnetosphere and magnetosheath plasmas, the first important ingredient to consider is the strong velocity shear that arises at the boundary, in addition to the magnetic shear which is the defining property of the magnetopause. Furthermore, the gradients concerning the particles and those concerning the magnetic field





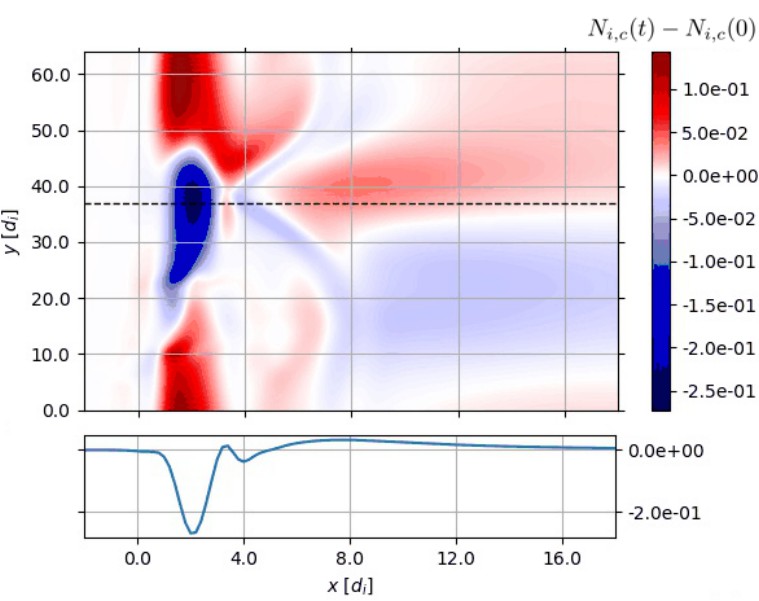

**Figure 6.** Shaded iso-contours of the cold ion fluctuations, $N_{i,c} - N_{i,c}(t=0)$ at $t = 1455\Omega_i^{-1}$. The bottom panel shows a plot of the same quantities $vs$ $x$ at $y \simeq 39$ corresponding to the horizontal dashed line in the shaded iso-contours. Numerical values are normalized to the magnetosheath density $N_{MSh} \sim 10cm^{-3}$.

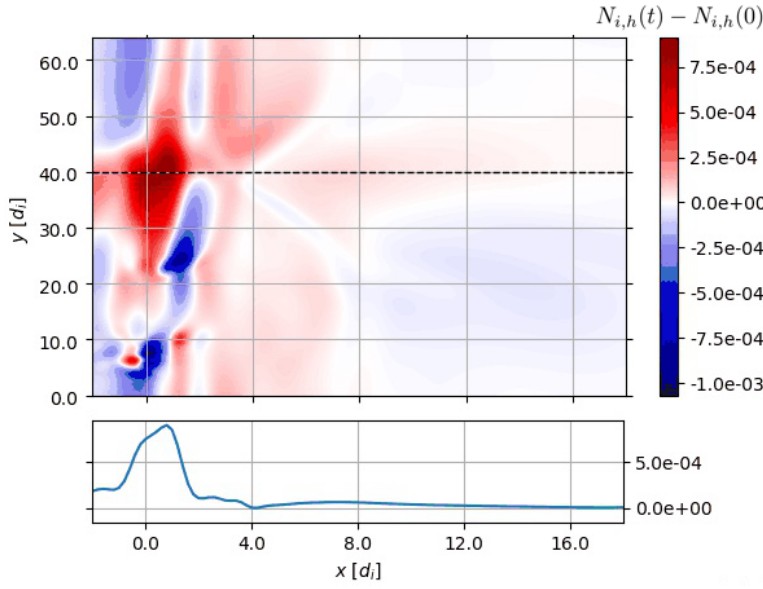

**Figure 7.** Same as Figure (6) for the hot ion fluctuations, $N_{i,h} - N_{i,h}(t=0)$.



most generally have different locations and show different scale-lengths. The model has also to be able to be take into account these characteristics.

In this paper, we present for the first time a three-fluid equilibrium directly derived from data, using a magnetopause crossing by MMS. The derivation of the model is based on a fit of the experimental data for the most reliable ones, completed by
a "realistic" solution of the equilibrium fluid equations for the others. The relative densities of the hot and cold ion populations calculated using the equilibrium equations provide an *a posteriori* check of our 3 fluid model. In particular, it helps understanding the different bulks observed on the ion distribution functions (see panel (d) in Figure (1)).

Furthermore, a preliminary study shows that the model can be implemented in a three-fluid numerical simulation, validating the correctness of the equilibrium solution. The detailed study of the long time evolution of the magnetopause instability will
be the subject for future work.

Investigating the magnetopause stability and trying to understand, in particular, when and where reconnection phenomena can be triggered and how the plasmas of both sides can get mixed, is still nowadays a challenging issue to be attacked by numerical simulations. However, known that the stability of a physical system is given by the specific initial equilibrium state, it must be kept in mind that the resulting non linear dynamics, in particular the mixing properties, also strongly depend on
the choice of the initial equilibrium. As a consequence it is very important to initialize a simulation with a configuration as much as realistic as possible. In most of the published literature, the simulations have been initialized with relatively simple configurations, Harris sheets, or modified Harris sheets with little relationship with the real magnetopause. The realistic 3fluid equilibrium presented in this paper should therefore allow for a step further. The same method could be applied to other experimental cases in the future.

*Data availability.* All the data used are available on the MMS data server: https://lasp.colorado.edu/mms/sdc/public/about/browse-wrapper/.

*Competing interests.* The authors declare that no competing interests are present.

*Acknowledgements.* This project (FC) has received funding from the European Union's Horizon 2020 research and innovation program under grant agreement No 776262 (AIDA). The French involvement on MMS is supported by CNES and CNRS.



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
