# Peer review of "A multi-fluid model of the magnetopause"

_Annales Geophysicae, 2019_

## Referee Comment (RC1) · Anonymous Referee #1 · 19 Sep 2019

Overall appreciation

This paper discusses a new multi-fluid model of the magnetopause as well as how it can be used to initiate a time-dependent simulation to verify magnetopause stability. The model is first used to represent a specific high magnetic shear magnetopause crossing observed by MMS. A time-dependent version of the model is then used to study the time evolution of the observed configuration. The paper is well-structured, and the reasoning and conclusions are clear overall, although several aspects should be explained better. The title is rather generic and does not fully do justice to the contents: It should reflect both the fitting of an equilibrium tangential discontinuity solution to the MMS data and the study of the time-dependent evolution of the structure. I do have some problems with how the paper sketches the broader context of magnetopause models in the introduction (see below). The language is mostly clear; a list of minor suggestions and typos is given below. This work appears to be intrinsically

sound. The questions that I want to raise here mostly have to do with the presentation of the material. The paper therefore will likely be suitable for publication after revision.

Questions and remarks

The paper considers the magnetopause interface between magnetosheath and magnetospheric plasmas. It does so by distinguishing a hot and a cold ion population (magnetospheric and magnetosheath, respectively) and a neutralizing electron population. The choice of the ions suggests that one is dealing with two plasmas from two different origins. And indeed, also the electron populations on the magnetospheric and magnetosheath sides are observed to have significantly different temperatures and form two different populations. So how does the assumption of a single electron population limit the applicability of the model to the magnetopause situation? The authors should discuss this issue in some depth, for instance at lines 35-38, where they argue for a three-fluid model. The question that should be answered is: why not a four-fluid model (magnetosheath and magnetospheric electrons + magnetosheath and magnetospheric ions)?

The introduction offers a discussion of magnetopause models. This discussion should be more clear in distinguishing magnetopause models and models for plasma entry. On line 44, there is a statement regarding the electric field, which is in general not correct. Tangential discontinuity models self-consistently compute the electric field, so imposing an initial electric field will constrain, for instance, the plasma parameters (see e.g. Roth et al. 1996, https://doi.org/10.1007/BF00197842). I do not see why steady-state models with distribution functions based on conserved quantities could not be considered "multi-population models"; they actually do include multiple populations. One may view these models as representing the end result of a time evolution. This has been discussed at length by Whipple et al., 1984, https://doi.org/10.1029/JA089iA03p01508. It is not because the accessibility problem is not solved in these models that it is not considered (for instance via the boundary conditions). And in fact, this is not so much due to the fact that the models work with conserved quantities, but with the fact that

they address tangential discontinuities. Note that there also have been some mixed approaches that resolve the accessibility issue (at least partially) for rotational discontinuities (such as Lee and Kan 1982 https://doi.org/10.1029/JA087iA01p00139).

Regarding existing magnetopause models, the authors paint an overly pessimistic portrait. Some magnetopause models, despite their simplifying mathematical assumptions, do go quite a long way in representing physical properties of such plasma boundaries. The equilibria they provide are not that unrealistic. Even though these models may have a number of free parameters, many of the conclusions that they arrive at are fairly robust, e.g. about boundary thickness (Grad, 1961 https://doi.org/10.1063/1.1706226), about the maximum possible tangential flow shear and the sense of magnetic field rotation (De Keyser & Roth, 1998 https://doi.org/10.1029/97JA03710), the presence of a density maximum at the high-magnetic-shear magnetopause (Hubert et al., 1998, https://doi.org/10.1029/97JA03298), etc.

The use of the model to study the time evolution of the observed structure poses a few questions that should be clarified. One of the main findings is that the observed structure is unstable and would immediately develop reconnection features. In that case, the observed structure would be out of equilibrium, so how can it then be used as a basis for examining magnetopause stability? Also, as the authors indicate, the structure would immediately lead to reconnection, i.e. one would obtain a rotational discontinuity. How probable is it then that – at the moment MMS passes this structure – it would still be of a tangential discontinuity nature, as the authors have identified earlier on?

Minor issues

14: Magnetosphere -> magnetosphere

22: Kelvin-Helmholtz -> Kelvin-Helmholtz instability

23: entering -> entry

29: It is a bit strange to cite a paper from 1998 (before the Cluster launch) to show that space plasma modelling has advanced during the Cluster era . . .

40: "Such models": It would be illuminating to cite some examples to make clear what models are meant here.

43: Models that consider a plane tangential discontinuity do not model plasma exchange (there is no net mass transport in such models), so these do not correspond to what is mentioned on line 40.

44: When talking about tangential discontinuities: yes, the electric field can be computed self-consistently from the plasma parameters, or, conversely, given the electric field profile there are constraints on the plasma parameters.

49: "let say" -> "let's say" or drop it altogether

49: all the other -> all the others

64: As it will be shown -> As will be shown

65: provides a -> provides

68: Please state at the beginning where this magnetopause crossing takes place – at the flanks? Near the subsolar point?

81: individuate -> distinguish

82: In this region of space, one finds both particles of magnetospheric and magnetosheath origin. In phase space, however, these particles do not occupy the same regions. Therefore, the word "mixing" might not be fully appropriate.

109 and 141: It might be useful to add a justification of the assumption of isotropic pressure and/or constant temperature for each population. In general, it is known that in a tangential discontinuity between species with isotropic Maxwellian distributions the par-

tial pressures become anisotropic inside the discontinuity layer (see, e.g. several figures with tangential discontinuity equilibria, as well as the expressions for temperature in Appendix A, in the review by Roth et al., 1996, https://doi.org/10.1007/BF00197842).

117: As written here, equation 1a and 1b assume that only electrons and singly charged ions are present; this is too restrictive if one would like to include the magnetosheath alpha particles.

122: Bad sentence construction or missing punctuation marks

Figure 2: It might be instructive to replace this plot by one that gives four curves: Ptot (constant), Pmag, Pion, and Pe. That would not only show that Pe is always much smaller, but would also illustrate how Pion and Pmag compare.

149: drop the comma

152: electrons frame -> "electron frame" or "electrons' frame"

182-183: This is not new and may differ depending on the situation and on how one defines the position of the plasma density transition; it should not be over-emphasized. The different thicknesses and positions of the magnetic field and plasma transitions at the magnetopause have been discussed and demonstrated repeatedly from an observational point of view (e.g. De Keyser et al., 2005, https://doi.org/10.5194/angeo-23-1355-2005) but also in the context of kinetic magnetopause models (e.g. De Keyser et al., 2017, https://doi.org/10.1002/9781119216346.ch7).

190-193: As already indicated before, a temperature discrepancy is not surprising since temperature anisotropies automatically appear due to the non-Maxwellian nature of the particle distributions inside the transition layer (Roth et al. 1996 https://doi.org/10.1007/BF00197842). To immediately claim that the observed structure is out of equilibrium seems rather drastic; the authors then should have selected another magnetopause crossing example in the first place.

197: It would be good to make clear which $d_i$ is used here. Moreover, it would be

illuminating if the length scales for magnetosheath and magnetospheric ions would be expressed in terms of their own gyroradii.

202 and Figure 3d: Explain why the presence of a normal electric field component is not in disagreement with the exact plasma neutrality condition of equation 1a.

Figures 3f and 3g: How are the observed currents determined? Is that done using the curlometer? Please provide some more details either in the caption or in the main text.

Figure 3: Please also add a plot for Te (at least for the model result)

222: observe -> observes

299: the defining property -> a defining property

232: I would suggest ". . . that solves the time-dependent set of equations . . ." since the authors earlier only used those equations with d/dt = 0.

307: bulks -> bulk quantities

308: Again, I suggest "in a time-dependent three-fluid numerical simulation"

313: known -> knowing

315-316: drop "as much"

Varia: Please be consistent with the use of "three-fluid", "three fluid", "3fluid", "2fluid" . . .

---

## Author Comment (AC1) · 1 Oct 2019

The comment was uploaded in the form of a supplement:
https://www.ann-geophys-discuss.net/angeo-2019-135/angeo-2019-135-AC1-supplement.pdf

---

## Author Comment (AC2) · 1 Oct 2019

We thank Referee 1 for his/her comments and useful remarks. In the following we include our answers point-by-point.

1) *The title is rather generic and does not fully do justice to the contents:*
*It should reflect both the fitting of an equilibrium tangential discontinuity solution*
*to the MMS data and the study of the time-dependent evolution of the structure*

Our choice intentionally put the accent on the question of "how building a realistic multi-model of the magnetopause" rather than on the study of the magnetopause stability (and the consequent mixing issues) resulting from a specific initialization. In our mind, this is the real point we want to address. The stability study, as outlined in the text, is more an example or a case study of our main aim . A possible alternative for the title could be "Building a realistic model of the magnetopause for initiating a numerical simulation", even if we would prefer to maintain our first one.

2) *...So how does the assumption of a single electron population limit the applicability*
*of the model to the magnetopause situation? The authors should discuss this issue in*
*some depth, for instance at lines 35-38, where they argue for a three-fluid model. The*
*question that should be answered is: why not a four-fluid model (magnetosheath and*
*magnetospheric electrons + magnetosheath and magnetospheric ions)?*

This remark is pertinent and it was a serious oversight not to have mentioned it in the text. Our final goal is indeed to build up a four-population model capable of distinguishing the magnetosheath and magnetospheric electron populations. The present paper must be considered as a first step in this direction. The main reason for starting with only one electron population was to avoid, as a first step, to enter into too many details for these species that are not likely to have a major role in the equilibrium. Actually, to model correctly the electrons in the magnetopause vicinity, it would be not sufficient to distinguish between one single magnetosheath population and one single magnetospheric one. In particular one should split the magnetospheric electron population itself into at least two sub-populations: one "cold" (poorly measured), carrying the density, and one "hot" carrying the pressure. Last but not least, we must say that starting from a previous already existing two-fluid code, it has been not so difficult to build up a code with several ion populations under the same quasi-neutral hypothesis, the electrons just providing an "Ohm's law". On the contrary, building up a fully multi-population model with several electron populations requires a radically different algorithm. We have actually developed this new algorithm, but it still remains to be implemented and tested. All these explanations will be added in the revised version.

3) *Regarding existing magnetopause models, the authors paint an overly pessimistic*
*portrait. Some magnetopause models, ....*

In our paper we have included a paragraph introducing the general context and briefly summarizing the state of the art of the most relevant models

available in the literature. Nevertheless, this paragraph is relatively short being not the central topic of the paper. We admit that our summary was too short, in particular when addressing the kinetic models. For these reasons, in the revised version we will modify this paragraph in order to avoid all possible ambiguous claims that could lead to any misunderstanding about this point. Keeping our willingness to be short, and without suppressing completely this part about kinetic equilibria, we now cite *Whipple et al* about the "accessibility problem", which is related to the confinement of the particles within their Larmor radius. We now also distinguish between models based on the invariant conservation only (Channel: one single function of the invariants assumed valid everywhere for f(v)), those partly introducing the accessibility problem (Roth et al: two different functions for the two sides), and those introducing it more completely (Belmont et al, Dorville et al). The last ones make the natural transition between the kinetic and MHD equilibria, the thickness of the magnetopause being not imposed a priori to be equal to the thermal Larmor radius (in MHD, the accessibility problem is extreme since each particle is confined within an infinitely small Larmor radius). However, we would like to maintain our point that the normal electric field, as shown in Belmont and Dorville, is not in general determined in the kinetic models of tangential layers. Let us recall for instance that it is assumed null in the most classical one: the Harris model. In other words, the normal electric field can be a byproduct of the model, but it is then a consequence of the simplifying assumptions of the model itself and not imposed by the Vlasov-Maxwell system of equations.

4) "The use of the model to study the time evolution of the observed structure poses a few questions that should be clarified..."

We agree with the referee's remark that it may appear contradictory to consider the data as characteristic of some magnetopause equilibrium and observe afterward that this equilibrium is not stable and should not last for long (even if the reconnection phenomenon is not "immediate"). To justify this point, we argue that the main characteristics which are taken into account are the asymptotic values on each side and the velocity shear between magnetosheath and magnetosphere. These conditions are not changed by the instability. The positions and the scale of the different gradients can indeed be partly modified by the instability, but we think that this is one of the interesting issues that can be investigated by the time evolution observed in the simulation. How the system stability is impacted? (need of a parametric study); how does it change in time due to non linear effects?; will the simulation tend toward a new more stable equilibrium state? This is left for further work. We will try to make it clearer in the text paper.

5) Minor issues. We thank the referee for his careful reading. We will fix all these points in the final version following your suggestions.

---

## Referee Comment (RC2) · Anonymous Referee #2 · 20 Oct 2019

**1 General comments:**

The paper presents a newly-developed magnetopause profile model, obtained by a novel and interesting combination of analytic theory and observation quantities. There is a clear need for models such as the presented one, since the only other sources of information about the magnetosphere have shortcomings: satellite measurements are limited to the points in which they were taken and can not easily be generalized, while global kinetic simulations are still numerically very expensive and can thus only be run for a limited number of cases.

While the structure of the paper is clear and straightforward, first constructing the theoretical model and then presenting numerical verification, there are some problems with how the two are connected:

[Figure]

I don't understand how the simulation results in section 5 validates the equilibrium solution that is presented before. As the authors strictly focus their analysis on the reconnection instability (getting steadyly growing instability results, including a linear and an nonlinear phase), the only result they seem to get is that their equilibrium solution is not in any equilibrium at all. I am missing a quantitative investigation of how the initial profiles develop over time and a discussion of how their deviation from the ideal values calculated before places caveats on their usability.

**2  Specific comments:**

- Line 64  71 reference a Manuzzo et al 2019 paper, which is apparently under review and does not seem to be publically available. This makes it somewhat awkward to understand the precise nature of MMS data that is being compared against. I suggest giving a compact explanation of the method, if it is possible, so that the input data can be appraised while the referenced paper is still under review.

- Equation 1b) Why is only $\text{sign}(q)$ being used in the equation and not q itself? What is Nabla bar? Is this an unusual unit system of Maxwell's equations?

- Equation 2 / line 125: Is the $P_{tot}$ here assumed to be a constant over the entire box, or a spatially varying quantity in accordance to observations?

- Line 140: Likewise, is this a global constant, or a spatially varying one? Please clarify.

- Equation 8: This interpolation is described as being performed for each quantity of interest independently, and it seems to be implied that this includes the magnetic field components. However, if this is performed for each B component individually, does it maintain $\text{div})(\vec{B}) = 0$?

- Line 237: I do not understand what a "spectral like resolution" in a finite difference scheme is supposed to be. Do you refer to it's accuracy as being comparable to that of spectral solvers? If so, by which measure do you consider them to be "spectral like"?

- Line 239: Please explain the coordinate system. If this is a 2D code, why are there x,y and z coordinates?

- Equation 9: The choice of epsilon is confusing here. Make sure to give it more distinction to the epsilons used before.

- Equation 9: what are the quantities i and j, mentioned as $i \neq j$ in this equation set?

- Figure 4 should have axes units or at least explanatory references in it's caption, as in it's current form it is not understandable without reading referenced literature.

- Figure 4 and 5c should reference each other, or might even be overplotted in the same axis.

- Figure 6: If the numerical values are normalized to $N_{MSh}$, why isn't this reflected in the colorbar unit label?

**3  Conclusion:**

The presented manuscript provides a novel and interesting magnetosphere model. However, it's current presentation is not entirely convincing and will require some revision.

---

## Author Comment (AC3) · 16 Nov 2019

**ANSWER TO REFEREE 1**
**(ADDENDUM TO FIRST ANSWER)**

We thank the Referee for her/his interesting and encouraging comments. Here some new answers to her/his remarks which must be considered as an addendum to the first response already sent to the Referee on October 1st.

**Title**. We have taken into consideration the Referee remark. However, we have finally decided to keep our initial title since we want to put the accent on the question of "how building a realistic multi-fluid model of the magnetopause" rather than on the study of the magnetopause stability (and related mixing issues) resulting from a specific initialization. The simulation presented is just a first proof a feasibility and, concerning the magnetopause stability, the study remains here preliminary. Nevertheless, we are open to consider new remarks, if needed.

**Why not two electron populations**?
We thank the referee for raising this point that we missed in the text. A new paragraph is now added in the Introduction.

**Kinetic magnetopause models**. We have re-organized the whole paragraph in the Introduction in order to better focus the argument.

**Why fitting the data to build an equilibrium since we are not sure that the observations are those of an equilibrium**? This is certainly a crucial- point. We have tried to discuss this point in a clearer way (see conclusion).

**Minor points**. All minor points reported by the referee have been fixed and we are grateful to the Referee for her/his suggestions.

**It is a bit strange to cite a paper from 1998 (before the Cluster launch) to show that space plasma modelling has advanced during the Cluster era**. In our opinion it is not so strange since a huge work has been done before the launch to prepare the analysis of the data. On the top of that we note that a first launch has failed before (1995).

**In this region of space, one finds both particles of magnetospheric and magnetosheath origin**. In phase space, however, these particles do not occupy the same regions. Anyway, there is no way to distinguish the origin of a particle either in real or in phase space.

**As written here, equation 1a and 1b assume that only electrons and singly charged ions are present; this is too restrictive if one would like to include the magnetosheath alpha particles**. The aim of the study is not to consider the role of the alpha particles, but the role of hot and cold protons. We have changed ions into protons at the beginning of Section 3.1.

---

## Author Comment (AC4) · 16 Nov 2019

The Revised manuscript is in pdf format

Please also note the supplement to this comment:
https://www.ann-geophys-discuss.net/angeo-2019-135/angeo-2019-135-AC4-supplement.pdf

---

## Author Comment (AC5) · 16 Nov 2019

**ANSWER TO REFEREE 2**

We thank the referee for her/his appropriate and encouraging comments which helped us to improve the quality of the paper. In the following our answers. In the re-submitted version of the paper all changes are in red.

- *The paper presents a newly-developed magnetopause profile model, obtained by a novel and interesting combination of analytic theory and observation quantities. There is a clear need for models such as the presented one, since the only other sources of information about the magnetosphere have shortcomings: satellite measurements are limited to the points in which they were taken and cannot easily be generalized, while global kinetic simulations are still numerically very expensive and can thus only be run for a limited number of cases. While the structure of the paper is clear and straightforward, first constructing the theoretical model and then presenting numerical verification, there are some problems with how the two are connected: I don't understand how the simulation results in section 5 validates the equilibrium solution that is presented before. As the authors strictly focus their analysis on the reconnection instability (getting steadly growing instability results, including a linear and a nonlinear phase), the only result they seem to get is that their equilibrium solution is not in any equilibrium at all. I am missing a quantitative investigation of how the initial profiles develop over time and a discussion of how their deviation from the ideal values calculated before places caveats on their usability.*

Concerning the distinction between an unstable equilibrium and the absence of equilibrium, we have slightly rearranged this part to show more clearly that we have tested the two properties. To summarize here this discussion, in the absence of an initial perturbation everything remains steady in the simulation (except for the numerical noise which can be easily controlled being about many order of magnitude smaller) until a time larger than one thousands of characteristic dynamical times, so proving that our model equilibrium is indeed so. On the other hand, when we add an initial perturbation to our model equilibrium the reconnection instability develops.

- *Line 64 71) reference a Manuzzo et al 2019 paper, which is apparently under review and does not seem to be publically available. This makes it somewhat awkward to understand the precise nature of MMS data that is being compared against. I suggest giving a compact explanation of the method, if it is possible, so that the input data can be appraised while the referenced paper is still under review*

We agree with the Referee about the need of briefly introducing the new technique developed in Manuzzo et al 2019. This is now discussed at the beginning of Section 2 and the paper is now accepted for publication. It will be soon available online.

- *Equation 1b) Why is only sign(q) being used in the equation and not q itself? What is Nabla bar? Is this an unusual unit system of Maxwell's equations?*

We agree with the Referee that an adimensional form of the starting equations, as we did, could lead to misunderstandings. For this reason the system of equations (1a)-(1g) is now re-written in dimensional form (SI units). On the other hand, for computational reasons, an adimensional form is used for the numerical code. Finally, nabla bar was a mistake, it is the standard nabla symbol (now fixed everywhere).

- *Equation 2 / line 125: Is the Ptot here assumed to be a constant over the entire box, or a spatially varying quantity in accordance to observations?*

In a 1D equilibrium model this quantity must be spatially constant. We have slightly rearranged this part to make it clearer.

- *Line 140: Likewise, is this a global constant, or a spatially varying one? Please clarify.*

  The clarification has been added to the text.

- *Equation 8: This interpolation is described as being performed for each quantity of interest independently, and it seems to be implied that this includes the magnetic field components. However, if this is performed for each B component individually, does it maintain div(**B**) = 0?*

  The equation div(**B**) = 0 is verified since the model is 1D with variations along n only, and with $B_n=0$. The fitting of **B** concerns the tangential components only.

- Line 237: I do not understand what a "spectral like resolution" in a finite difference scheme is supposed to be. Do you refer to it's accuracy as being comparable to that of spectral solvers? If so, by which measure do you consider them to be "spectral like"?

  Spectral like resolution is the name given by Lele, JCP (1992) to the possibility of build up an implicit finite difference operator (i.e. including the nearby values of the derivatives) asking not only to minimize at max the accuracy (Taylor development) but also to solve the most possible Fourier equivalent wave vectors, so a mixture between finite differences and spectral methods. In the text we now explicitly refer to Lele (1992) for the significance and technical details of compact finite differences.

- *Line 239: Please explain the coordinate system. If this is a 2D code, why are there x,y and z coordinates?*

  We thank the reviewer for raising this point. Now any reference to the z direction has been deleted. For the sake of simplicity in this work we limit to a 2D geometry, but the numerical code is fully 3D. This sentence has been added to the text.

- *Equation 9: The choice of epsilon is confusing here. Make sure to give it more distinction to the epsilons used before.*

  We thank the Referee for letting us fix this misprint. We have now re-defined the perturbation amplitude with the symbol *a* both in Eq. 9 and 10.

- *Equation 9: what are the quantities i and j, mentioned as i 6= j in this equation set?*

  The question is no longer relevant following the modification of the equation.

- *Figure 4 should have axes units or at least explanatory references in it's caption, as in it's current form it is not understandable without reading referenced literature.*

  The axes have adimensional units (added in the caption).

- *Figure 4 and 5c should reference each other, or might even be overplotted in the same axis.*

  Correction made in the text (line 308-309)

- *Figure 6: If the numerical values are normalized to NMSh, why isn't this reflected in the colorbar unit label?*

  The units have been added to both figure 6 and 7.

---

## Referee Report (RR1)

**Referee report for revised paper "A multi fluid model of the magnetopause", Manuzzo et. al**

**December 29, 2019**

The authors have addressed my comments from the first round of review comments and the manuscript has improved in legibility considerably. Especially in relation to my overarching comment that the connection between the analytic model and discussion of simulation results seemed to be disconnected from one another, the authors have made considerable modifications to the sections motivating their simulation approach and discussing the importance of the obtained results.

The addition of the sentence in line 272, where the long term stability of the unperturbed configuration in the numerical code is explicitly highlighted, further helped to underline the versatility of the chosen approach.

I still have some minor comments remaining for the manuscript before I can recommend it for publication:

- While the introduction mentions kinetic simulations as a potential source for magnetopause equilibrium data, the scope of referenced literature is quite narrow. The recent advent of global kinetic and hybrid-kinetic simulation models such as Chen et al. (2018) ( `https://doi.org/10.1002/2017JA024186` ) Karimabadi et al. (2014) ( `https://doi.org/10.1063/1.4882875` ) or Palmroth et al. (2018) ( `https://doi.org/10.1007/s41115-018-0003-2` ) provides an additional avenue to analyse and provide magnetosphere equilibria. Addition of a short discussion of their properties in relation to the manuscript at hand would be welcome.

- In the updated formulation of the multi-fluid equations (eq. 1a - 1g), some equations appear to use $\alpha$ as a species index, while others use $\beta$. The distinction between the two (if there is any) does not become clear from the text. Please clarify or homogenize the terminology used in these equations further.

- The model described in section 3 apparently works excellently for the described example case of a tangential magnetopause situation near GSE z=0, as its mathematical formulation is based on this assumption. I am still missing some discussion about the boundaries of applicability of the model for magnetopause locations further away from z=0, or further removed from the dayside reconnection. A short critical discussion of conditions that would make the presented model less reliable would help the reader appreciate its value in the domain of applicability.

Some notes about the referenced literature:

- Neither a URL nor DOI has been listed for the paper by Alvarez Laguna. Is it this one? `https://lirias.kuleuven.be/retrieve/531033`

- Please give an ISBN reference for the book by Canuto

- The papers by Dargent et al., Dungey at al. and Shumlak et al. are giving duplicate DOIs

- The papers be Lee et al. and Lele et al. are referenced using both DOI and URL, which is redundant.

- A DOI should be added for the paper of Modolo et al.

Altogether I believe the manuscript is presenting an important new result and can be accepted for publication once these minor comments are addressed.

---

## Author Response (AR2)

**ANSWER TO REFEREE 1**

We thank the Referee for his/her comments which helped us in improving the manuscript. All changes/adds in the manuscript are in red. The answers to the questions follow.

1. *On line 52-53 the authors state "Unfortunately such models are very complicated...*
2. *In the reviewer's response it is said that the electric field is a consequence of the simplifying*

The "two points" raised by referee 1 concern the kinetic equilibria. It seems that his/her concerns rely on a different perspective between one of the co-authors (Gerard Belmont) who has already published several papers on this subject, and the referee, who is certainly a specialist on it. This co-author will be very interested and happy to continue the debate with the referee later on. On the other hand, as the paper is concerned,  since this question has no direct influence on the content of the paper itself, we gladly accept to suppress the disputed sentences in order to avoid delaying in the publication. Therefore, we have replaced them in the text by purely factual sentences that are, hopefully, not disputable.

**ANSWER TO REFEREE 2**

We thank the Referee for his/her comments which helped us in improving the manuscript. All changes/adds in the manuscript are in red. The answers to the questions follow.

1. *While the introduction mentions kinetic simulations...*

   We have inserted a new sentence at line 65 to discuss the global simulations approach.

2. *In the updated formulation of the multi-fluid equations ...*

   The definition of indices alpha and beta has been added in the text.

3. *The model described in section 3 apparently works excellently ...*

   The 3-fluid model is not related to the place where the observations are made. It can be applied to any crossing of the magnetopause.

4. *Some notes about the referenced literature:*

   Thanks. All misprints have been fixed.